# Transcriptome Analysis in Response to Infection of *Xanthomonas oryzae* pv. *oryzicola* Strains with Different Pathogenicity

**DOI:** 10.3390/ijms24010014

**Published:** 2022-12-20

**Authors:** Min Tang, Hui Zhang, Yao Wan, Ziqiu Deng, Xuemei Qin, Rongbai Li, Fang Liu

**Affiliations:** State Key Laboratory for Conservation and Utilization of Subtropical Agro-Bioresources, College of Agriculture, Guangxi University, Nanning 530004, China

**Keywords:** rice, bacterial leaf streak (BLS), transcriptome sequencing

## Abstract

Bacterial leaf streak (BLS) caused by *Xanthomonas oryzae* pv. *oryzicola* (*Xoc*) is one of the most important quarantine diseases in the world. Breeding disease-resistant varieties can solve the problem of prevention and treatment of BLS from the source. The discovery of the molecular mechanism of resistance is an important driving force for breeding resistant varieties. In this study, a BLS-resistant near isogenic line NIL-*bls2* was used as the material. Guangxi *Xoc* strain gx01 (abbreviated as WT) and its mutant strain (abbreviated as MT) with a knockout type III effectors (T3Es) gene were used to infect rice material NIL-*bls2*. The molecular interaction mechanism of rice resist near isogenic lines in response to infection by different pathogenic strains was analyzed by transcriptome sequencing. The results showed that there were 415, 134 and 150 differentially expressed genes (DEGs) between the WT group and the MT group at 12, 24 and 48 h of post inoculation (hpi). Through GO and KEGG enrichment analysis, it was found that, compared with non-pathogenic strains, the T3Es secreted by pathogenic strains inhibited the signal transduction pathway mediated by ethylene (ET), jasmonic acid (JA) and salicylic acid (SA), and the MAPKK (MAPK kinase) and MAPKKK (MAPK kinase kinase) in the MAPK (mitogen-activated protein kinase) cascade reaction, which prevented plants from sensing extracellular stimuli in time and starting the intracellular immune defense mechanism; and inhibited the synthesis of lignin and diterpenoid phytochemicals to prevent plants from establishing their own physical barriers to resist the invasion of pathogenic bacteria. The inhibitory effect was the strongest at 12 h, and gradually weakened at 24 h and 48 h. To cope with the invasion of pathogenic bacteria, rice NIL-*bls2* material can promote wound healing by promoting the synthesis of traumatic acid at 12 h; at 24 h, hydrogen peroxide was degraded by dioxygenase, which reduced and eliminated the attack of reactive oxygen species on plant membrane lipids; and at 48 h, rice NIL-*bls2* material can resist the invasion of pathogenic bacteria by promoting the synthesis of lignin, disease-resistant proteins, monoterpene antibacterial substances, indole alkaloids and other substances. Through transcriptome sequencing analysis, the molecular interaction mechanism of rice resistance near isogenic lines in response to infection by different pathogenic strains was expounded, and 5 genes, *Os01g0719300*, *Os02g0513100*, *Os03g0122300*, *Os04g0301500*, and *Os10g0575100* closely related to BLS, were screened. Our work provides new data resources and a theoretical basis for exploring the infection mechanism of *Xoc* strain gx01 and the resistance mechanism of resistance gene *bls2*.

## 1. Introduction

BLS is one of the most destructive quarantine diseases in the world. It has the characteristics of early onset, rapid spread and strong destructiveness in the growth cycle. The rice yield can be reduced by 40~60% in areas with severe disease [1]. *Xoc*, the pathogen of BLS, enters rice leaves through stomata or damaged parts of leaves, colonizes intercellular spaces (apoplasts) of epithelial cells, spreads to other parts of plants through xylem vessels, infects parenchyma cells, and expands outward along the vein direction. In the early stage, the diseased spots are translucent and waterlogged, and in the later stage, the whole leaves are yellow and translucent streaks are visible to light [2].

Faced with invasion by pathogenic bacteria, plants have two immune defense mechanisms: pathogen-associated molecular patterns (PAMP), such as flagellin, which are recognized by the corresponding pattern recognition receptor (PRR) of plants, triggering broad-spectrum immune defense response PTI (PAMP-triggered immunity), and inhibiting the initial infection of pathogenic bacteria. In plant cells, specific effector factors of pathogenic bacteria are directly or indirectly identified by disease-resistant genes, which trigger the plant’s own hypersensitive response (HR), causing local cell death and preventing the further spread of pathogenic bacteria. This reaction is known as effector-triggered immunity (ETI) [3,4]. The PTI reaction gives plants long-lasting and broad-spectrum resistance, while the ETI reaction is mediated by polymorphic disease-resistant proteins, which has strong specificity. In order to better defend against the invasion of pathogenic bacteria, PTI and ETI reactions in plants usually occur simultaneously, causing various defense-related reactions: changes in signal pathways; synthesis of antibacterial secondary metabolites such as lignin [5,6], chalcone [5,7], phytoalexin [8] and alkaloid [9]; ubiquitin in protein [10], etc.

When *Xoc* infects rice, T3Es are mainly injected into host cells by a type III secretion system, thus overcoming the PTI immune response of plants and promoting bacterial growth. Plant resistance proteins trigger an ETI-specific immune response by recognizing T3Es, thus triggering an HR response [4]. This plant-pathogen interaction mechanism is the main focus of research on molecular mechanisms related to BLS resistance in rice at present. Studies have shown that regulating the expression of defense-related genes (DR) can improve the resistance of rice to *Xoc*. Overexpression of some DR genes can improve rice BLS resistance, genes such as *RGA4/RGA5* [11], *OsPGIP4* [12], *MAPK10.2* [13], *OsPSKR1* [14], *OsHSP18.0-CI* [15] and *GH3-2* [16]. The BLS resistance of rice was improved by inhibiting the expression of some DR genes, such as *OsWRKY45-1* [17], *OsDEPG1* [18], *OsNRRB* [19] and *OsMPK6* [20]. Recently, a recessive resistance gene *bls1* has been cloned, which encodes a mitogen-activated protein kinase (OsMAPK6). Both overexpression of *BLS1* and low expression of *bls1* could increase salicylic acid and induce the expression of defense-related genes, simultaneously increasing the non-race specific, broad-spectrum resistance. Moreover, low expression of *bls1* showed increase in JA and ABA (abscisic acid), in company with an increase the race specific resistance to the *Xoc* strain JZ-8 [21]. It can be seen that the defense system of rice against *Xoc* infection is complex, and the research on the molecular mechanism of rice resistance to BLS is at the initial stage, which needs to be further explored.

Transcriptome sequencing has become an effective method to study the gene regulation and signal network of plant defense responses [22], which not only promotes the genetic analysis and molecular breeding of rice, but also provides a basis for the development and utilization of rice genome resources. In our study, the molecular mechanism of NIL-*bls2* in response to different pathogenic *Xoc* strains was studied by transcriptome sequencing analysis, including related metabolic pathways and molecular regulatory networks, to provide a theoretical basis for revealing the interaction mechanism between rice and *Xoc*.

## 2. Results

### 2.1. Resistance Identification Results

NIL-*bls2*, a disease-resistant near isogenic line material of rice BLS, was inoculated with three treatments: the WT group was inoculated with pathogenic strain gx01, the MT group was inoculated with non-pathogenic gx01 mutant strain and the H_2_O group was inoculated with sterile H_2_O as a control group. The results of spot identification showed that there were no signs of disease in the H_2_O and MT groups. The average length of diseased spots in the WT group was 6.87 mm, showing BLS resistance (Figure 1).

### 2.2. RNA Sequencing Results

After three kinds of inoculation treatments (WT group, MT group and sterile water group) were carried out on rice material NIL-*bls2*, the samples of the three treatment groups were sampled at 12, 24 and 48 hpi. Each time point had three biological repeats. There were 27 samples sequenced on the illumina platform. The sequencing results met the analysis requirements: each sample has more than 6G of data, and the quality Q20 of sequencing data was more than 97%. Among the clean reads obtained through data filtering, the number of reads uniquely located in the reference genome was higher than 85%. We used these unique reads located in the reference genome for subsequent gene expression analysis (Appendix A Appendix A).

### 2.3. DEGs Identification Number Analysis

We removed the genes with no expression difference between the experimental group and the control group at each time point. The DEGs of the rice material in the WT and MT groups at different time points after inoculation were statistically analyzed to reveal the expression patterns of resistance-related genes. In rice NIL-*bls2*, 607 DEGs were identified between the WT group and MT group after different inoculation times (12, 24 and 48 hpi) (Appendix A). In the MT group, 415 DEGs (203 up- and 212 down-regulated), 134 DEGs (68 up- and 66 down-regulated), and 150 DEGs (69 up- and 81 down-regulated) were found at 12, 24 and 48 hpi, respectively. In contrast, 12 hpi samples had the largest number of DEGs. A Venn diagram shows the overlapping DEGs at different time points (Figure 2).

### 2.4. DEGs Resistance Gene Distribution

We carried out gene annotation on DEGs, a screening gene related to plant resistance. It can be divided into 4 categories (Appendix A): 10 transcription factor-related genes (MYB, MYC2, WRKY, ERF, bHLH), 16 protein kinase-related genes (WAK, CRK, LecRK), 64 antibody protein-related genes (NBS-LRR, LRR, pathogen-related, disease resistance, sugar transporter calmodulin-like, ubiquitin family, protein sulphate transporter, glucosyltransferase, cytochrome P450, PP2C, etc.) and 8 hormone-related genes (JA, ET, Gas). Among these genes related to plant resistance, the annotation information of 5 genes was closely related to BLS, and the differential expression was significant (Appendix A Yellow background content). The related genes are: sulfate transporter gene *Os01g0719300(OsSULTR3:6)*, sucrose transporter gene *Os02g0513100 (OsSWEET15)*, 2-oxoglutarate-dependent dioxygenase gene *Os03g0122300* (*OsF3H03g*), bHLH transcription activator regulate-related gene *Os04g0301500 (RERJ1; OsbHLH6)*, and transcription factor MYC2-related gene *Os10g0575100*.

### 2.5. GO Enrichment Analysis of DEGs

Among the 415 DEGs identified at 12 hpi, GO annotated 303 genes, including 144 up-regulated genes and 159 down-regulated genes. The second level of GO was enriched to 19 biological process items, 11 cellular component items and 10 molecular functions (Appendix A). The significance of all the enriched levels was analyzed (*p* < 0.01) (Figure 3a), (Appendix A). Among the biological process terms, the two most significant terms were: “reactive oxygen species process (GO:0072593)” (the majority of genes are down-regulated) and “hydrogen peroxide catabolic process (GO:0042744)” (all genes down-regulated). Among the cellular component terms, the prominent term was “extracellular component (GO:0005576)”. Among the molecular function terms, the five most significant terms were: “tetrapyrrole binding (GO:0046906)”, “oxidoreductase activity, acting on peroxide as acceptor (GO:0016684)” (all genes down-regulated), “antioxidant activity (GO:0016209)” (all genes down-regulated), “oxidoreductase activity (GO:0016491)” and “cation binding (GO:0043169)”. To sum up, compared with the non-pathogenic bacteria, when the pathogenic bacteria had infected the rice for 12 h, the changes of cell peripheral components in rice were more obvious, and the processes of active oxygen metabolism and hydrogen peroxide catabolism were more active, the related genes were down-regulated, which inhibited the metabolic process and led to the accumulation of active oxygen and hydrogen peroxide, caused membrane lipid peroxidation, and damaged the membrane system, which was beneficial to the invasion of bacteria. Antioxidant activity and oxidoreductase activity with hydrogen peroxide as receptor were more prominent, and related genes were down-regulated, so that their activities were inhibited, which led to the failure of plant defense system to prevent the invasion of pathogenic bacteria by reducing reactive oxygen species and decomposing hydrogen peroxide.

Among the 134 DEGs identified at 24 hpi, GO annotated 106 genes, including 56 up-regulated genes and 50 down-regulated genes. The second level of GO was enriched to 17 biological process items, 13 cellular component items and 6 molecular function items (Appendix A). Significance analysis was carried out on all levels of enrichment (*p* < 0.01) (Figure 3b), (Appendix A). The two prominent biological process-related terms were: “oxidation-reduction process (GO:0055114)” and “negative regulation of defense response to bacterium (GO:1900425)” (down-regulation of participating genes). Among the cellular component items, the prominent term was “cell periphery (GO:0071944)”. Among the molecular function terms, the prominent term was “dioxygenase activity (GO:0051213)” (all genes were down-regulated). To sum up, compared with non-pathogenic bacteria, when pathogenic bacteria had infected the rice for 24 h, the oxidation-reduction reaction, the defense reaction against bacteria and the reaction involving dioxygenase in rice were more active. The down-regulation of related genes involved in bacterial defense reaction was beneficial to the invasion of pathogenic bacteria. Most related genes involved in dioxygenase activity are up-regulated, which indicates that rice can degrade hydrogen peroxide by dioxygenase, and reduce and eliminate the attack of reactive oxygen species on plant membrane lipids.

Among the 150 DEGs identified at 48 hpi, GO annotated 128 genes, including 58 up-regulated genes and 70 down-regulated genes (Appendix A); The second level of GO was enriched to: 17 biological process items, 10 cellular component items and 7 molecular function items. Significance analysis was carried out on all the enriched levels (*p* < 0.01) (Figure 3c), (Appendix A). Among the biological process terms, the prominent term was “diterpenoid metabolic process (GO:0016101)” (all genes down-regulated). Among the molecular function terms, the two most significant terms were: “terpene synthase activity (GO:0010333)” (all genes down-regulated) and “carbon-oxygen lyase activity, acting on phosphates (GO:0016838)” (all genes down-regulated). Compared with non-pathogenic bacteria, when pathogenic bacteria infected the rice for 48 h, the biosynthesis process of diterpenes in rice was inhibited.

### 2.6. KEGG Enrichment Analysis of DEGs

According to the time point, the identified DEGs underwent KEGG metabolic pathway enrichment analysis. The KEGG pathway was mainly distributed in five categories: organic system, environmental information processing, cell process, metabolism and genetic information processing (Appendix A).

Among the 415 DEGs identified at 12 hpi, 112 DEGs were involved in 47 KEGG pathways, including 51 up-regulated genes and 61 down-regulated genes. KEGG enrichment analysis of “phenylpropanoid biosynthesis (ko00940)”, “zeatin biosynthesis (ko00908)”, “MAPK signaling pathway–plant (ko04016)”, “plant hormone signal transduction (ko04075)”, and “stilbenoid, diarylheptanoid and gingerol biosynthesis (ko00945)” were significant (*p* < 0.01) (Figure 4a). By analyzing the specific gene information involved in the related KEGG pathway (Appendix A), it was found that all genes in the phenylalanine biosynthesis pathway were down-regulated, indicating that lignin synthesis was inhibited, which was beneficial to the spread of pathogenic bacteria. In the MAPK signaling pathway of plants, the genes of the first-order phosphorylation in the defense reaction initiated by flg22 and H_2_O_2_, the wound reaction and root growth involved by jasmonic acid, and the plant defense reaction mediated by ethylene were all down-regulated, and most of the genes enriched in the plant hormone signaling pathway were down-regulated. It is speculated that the secretion-specific effector of wild-type pathogenic strains mainly inhibits PTI reaction of plants through these pathways, and helps the spread of pathogens in plants.

Among the 134 DEGs identified at 24 hpi, 41 DEGs were involved in 33 KEGG pathways, including 15 up-regulated genes and 26 down-regulated genes. KEGG enrichment analysis of “diterpenoid biosynthesis (ko00904)” and “benzoxazinoid biosynthesis (ko00402)” was significant (*p* < 0.01) (Figure 4b). By analyzing the information of genes involved in the KEGG pathway (Appendix A), it was found that the genes enriched by the diterpene biosynthesis pathway were down-regulated, which inhibited the synthesis of diterpene antitoxin, thus beneficial to the infection and proliferation of pathogenic bacteria in rice, The genes enriched by the benzoxazinoid biosynthesis pathway were up-regulated, and were beneficial to plant disease resistance.

Among the 150 DEGs identified at 48 hpi, 43 DEGs were involved in 31 KEGG pathways, including 17 up-regulated genes and 26 down-regulated genes. The KEGG enrichment analysis of “phenylpropanoid biosynthesis (ko00940)” and “diterpenoid biosynthesis (ko00904)” were significant (*p* < 0.01) (Figure 4c). By analyzing the information of genes involved in the KEGG pathway (Appendix A), we found that all genes enriched in diterpene biosynthesis pathway were still down-regulated, which inhibited their synthesis. The genes related to the shikimic acid core unit in phenylalanine biosynthesis pathway were still down-regulated, while those closely related to lignin synthesis were up-regulated, which indicated that the inhibitory effect of pathogenic bacteria on lignin synthesis was gradually weakened.

### 2.7. Key KEGG Pathway Analysis

The reported KEGG pathways related to plant disease resistance mechanism mainly focused on the phenylalanine biosynthesis pathway, the plant hormone signaling pathway, the plant MAPK signaling pathway, the plant-pathogen interaction pathway, the diterpene biosynthesis pathway and the flavonoid biosynthesis pathway [22,23,24,25,26,27].

In the plant-pathogen interaction pathway (Figure 5a), pathogenic bacteria inhibit the plant defense response by inhibiting the expression of the following genes. The related genes included calmodulin-related gene *Os01g0949260*, calcium-dependent protein kinase-related gene *Os03g0788500*, first-order phosphorylated MEKK1-related gene *Os03g0262150* and second-order phosphorylated MKK4/5-related gene *Os02g0787300* in cascade phosphorylation signal transduction and disease resistance protein RPM1-related genes *Os08g0424700*.

Among the plant hormone signal transduction pathway (Figure 5b) and the MAPK signal pathway (Figure 5c), pathogenic bacteria inhibited the signal transduction reaction from extracellular stimuli to intracellular stimuli by inhibiting the expression of related genes in the signal transduction pathway dependent on abscisic acid, ethylene and jasmonic acid, thus inhibiting the plant defense response. Related genes included the abscisic acid receptor PYR/PYL family-related gene *Os05g0473101* in the stomatal closure pathway mediated by abscisic acid; the serine/threonine protein kinase CTR1-related gene *Os02g0527600* in ethylene-mediated plant defense response pathway; the first-grade phosphorylated MEKK1 gene *Os03g0262150* and second-grade phosphorylated MKK4/5 gene *Os02g0787300*; jasmonic acid-mediated monoterpene biosynthesis, indole alkaloid synthesis or transcription factor MYC2-related gene *Os10g0575100* in aging stress response and disease resistance protein RP1-related gene *Os07g0127600*. The expression of genes *Os05g0102000* and *Os03g0225900* related to jasmonic acid synthesis in α-linolenic acid metabolic pathway was down-regulated, which inhibited jasmonic acid synthesis (Figure 5d).

In the phenylalanine biosynthesis pathway (Figure 5e), the expression of a large number of peroxidase-related genes in the lignin biosynthesis pathway, shikimic acid O-hydroxycinnamoyl transferase-related genes *Os04g0664600* and *Os11g0182200* in the shikimic acid core unit, and β-glucosidase-related gene *Os01g0813700* in the coumarin biosynthesis pathway were down-regulated, which indicated that pathogenic bacteria inhibited plant defense response by inhibiting the synthesis of lignin and coumarin. In the WT group samples at 48 hpi, most genes in the lignin synthesis pathway were up-regulated, and it was speculated that plant lignin synthesis gradually increased at 48 h.

In the diterpene biosynthesis pathway (Figure 5f) and the flavonoid biosynthesis pathway (chalcone biosynthesis pathway) (Figure 5g), 12, 24 and 48 hpi-enriched genes were mostly down-regulated genes, which indicated that pathogenic bacteria inhibited the plant defense response by inhibiting the synthesis of diterpene antitoxin, chalcone and other secondary metabolites related to plant resistance.

A few up-regulated and down-regulated genes were found in the KEGG pathway related to the plant disease resistance mechanism, which suggested that these genes might be closely related to the resistance of rice NIL-*bls2* materials. Among the metabolic pathways of α-linolenic acid (Figure 5d), the genes related to the traumatic acid synthesis pathway in the WT group samples at 12 hpi were up-regulated, indicating that plants were repairing the inoculation wounds at this time. In the plant-pathogen interaction pathway, the plant hormone signal transduction pathway and the MAPK signal pathway (Figure 5a–c), the gene *Os07g0129300* was enriched in the WT group samples at 48 hpi, which participated in the HR response induced by pathogenic bacteria secretion flg22 or H_2_O_2_ as the disease resistance protein RPM1 gene. At the same time, it also participated in the SA-mediated signal transduction pathway related to plant resistance as a pathogenesis-related protein RP1 gene. In the plant hormone signal transduction pathway and the MAPK signal pathway (Figure 5b,c), JA-mediated monoterpene biosynthesis, indole alkaloid biosynthesis and stress response, the related gene *Os10g0575100* was significantly down-regulated at 12 hpi and 24 hpi, and significantly up-regulated at 48 hpi. In the phenylalanine biosynthesis pathway (Figure 5e), most genes in the lignin biosynthesis pathway were up-regulated in the WT group samples at 48 hpi. According to the analysis, rice NIL-*bls2* material can promote wound healing for 12 h by promoting the synthesis of traumatic acid. Through promoting the synthesis of lignin, disease-resistant protein, monoterpene antibacterial substances, indole alkaloids and other substances for 48 h, it can resist the invasion of pathogenic bacteria.

### 2.8. qRT-PCR Confirms Gene Expression Profiles

According to the results of the RNA-seq analysis, 8 genes (*p* < 0.01) were screened for qRT-PCR verification. They were the sulfate transporter-related genes *Os01g0719300*; a gibberellin 2β-dioxygenase-related gene *Os04g0522500* in the diterpene biosynthesis pathway; transcription factor MYC2-related gene *Os10g0575100* in the JA-mediated defense response in the plant hormone and MAPK signal transduction pathway and 5 randomly selected genes *Os01g0950866*, *Os01g0294700*, *Os03g0171700*, *Os05g0551900* and *Os07g0643400*. The expression trend of qRT-PCR results is basically consistent with that of RNA-seq sequencing results, which indicates that RNA-seq sequencing results are reliable (Figure 6).

## 3. Discussion

BLS has contributed to significant yield losses over the last decade [14], but currently there is little reported about major gene resistance to BLS pathogenesis [21]. A Guangxi common wild rice material DY19, which contains BLS-resistant gene *bls2*, shows high resistance to BLS [28]. We used DY19 as donor parent to develop a near isogenic line NIL-*bls2*, thus providing an ideal material for studies on rice BLS resistance. In this study, 607 DEGs were identified in NIL-*bls2* by transcriptome analysis after inoculation with *Xoc* strain gx01 and its non-pathogenic mutant at different time points (12, 24, 48 hpi). GO and KEGG enrichment analyses were carried out in order to understand the molecular mechanism of interaction between NIL-*bls2* and different pathogenic *Xoc* strains.

Using DEGs gene annotation, we found 98 genes (transcription factors, protein kinases, antibody proteins, hormones and other related genes) closely related to plant resistance. According to the analysis of DEGs gene annotation and KEGG enrichment, it was found that 5 DEGs were closely related to resistance to BLS (Appendix A): sulfate transporter gene *Os01g0719300 (OsSULTR3:6)*, sucrose transporter gene *Os02g0513100 (OsSWEET15)*, 2-oxoglutarate-dependent dioxygenase gene *Os03g0122300 (OsF3H03g)*, bHLH transcription activator regulate-related gene *Os04g0301500 (RERJ1; OsbHLH6)*, and transcription factor MYC2-related gene *Os10g0575100*. Sulfate transporter gene *Os01g0719300 (OsSULTR3:6)* has been reported as a BLS-sensitive gene, which can be bound by TALEs of *Xoc* pathogenic bacteria. Editing the promoter region effector binding elements of this gene can improve BLS resistance [29]. In this study, *Os01g0719300* is a significantly up-regulated gene, which is consistent with the reported content. Sucrose transporter gene *Os02g0513100 (OsSWEET15)* has been proved to be the TALEs target secreted by *Xanthomonas oryzae* pv. *oryzae* (*Xoo*), and it is a bacterial blight-sensitive gene [30,31]. The expression of this gene is significantly different in this study. It is speculated that the gene *Os02g0513100* can also be combined by TALEs secreted by *Xoc* strain gx01, which affects the resistance to BLS. The 2-oxoglutarate-dependent dioxygenase gene *Os03g0122300(OsF3H03g)* has been proved to negatively regulate the resistance of rice to *Xoc* and *Xoo* by directly reducing SA-related defense [32]. In this study, *Os03g0122300* was up-regulated in the WT group at 24 hpi and 48 hpi, corresponding to the reported results. The transcription factor MYC2-related gene *Os10g0575100* was enriched in the JA-dependent signal transduction pathway, and it was significantly down-regulated at 12 and 24 hpi, and up-regulated at 48 hpi. In resting cells, the JAZ protein acts as a JA transcription inhibitor by binding with the MYC2 positive transcription regulator [33]. In the nucleus OsbHLH6 (Os04g0301500) interacts with OsMYC2, which is a key regulatory protein in the JA signaling pathway; OsbHLH6 also can prevent OsJAZ from inhibiting the transcription of OsMYC2, which is a negative regulatory protein of the JA signal, thus promoting the activation of the signal [34]. In this study, *Os10g0575100* was highly down-regulated at 12 hpi and 24 hpi, and *Os04g0301500* was significantly up-regulated at 24 hpi, which indicated that the inhibition of the JA pathway was broken and the signal transduction reaction dependent on JA was activated. It is speculated that the transcription factor MYC2-related gene *Os10g0575100* is an important gene of the jasmonic acid-dependent pathway, which may interact with the bHLH transcription factor *Os04g0301500* gene to activate JA signaling pathway.

Comprehensive analysis of GO and KEGG pathway enrichment results showed that At 12 hpi, most of the genes enriched in the KEGG pathway were down-regulated, and the related pathways included: the ET, JA, SA-dependent signal transduction pathway; the 1st and 2nd phosphorylation in MAPK signal transduction pathway; in the phenylalanine biosynthesis pathway, shikimic acid core unit and peroxidase participate in the branch of lignin synthesis; the JA synthesis in the α-linolenic acid metabolic pathway and Ca^2+^-mediated ROS and NO synthesis in the plant-pathogen interaction pathway. At 24 hpi, the down-regulated gene pathways included: the JA and SA-dependent signal transduction pathway, the 1st phosphorylation in MAPK signal transduction pathway; peroxidase is involved in lignin synthesis in the phenylalanine biosynthesis pathway, and Diene skeleton synthesis and rice hull ketone synthesis in the diterpene biosynthesis pathway. Up-regulated gene pathways included the metabolic pathways of ascorbic acid and hydrochloride. At 48 hpi, down-regulated gene pathways included: the 1st phosphorylation in MAPK signal transduction pathway; Diene skeleton synthesis and rice hull ketone synthesis in the diterpene biosynthesis pathway. Up-regulation gene pathways included: the branch of peroxidase involved in lignin synthesis in the phenylalanine biosynthesis pathway; synthesis of disease-resistant protein RP1 in the signal transduction pathway and the defense reaction involving JA. Based on the above analysis, it is speculated that the T3Es secreted by pathogenic bacteria can inhibit the signal transduction pathway dependent on ET, JA and SA and mediated by 1st and 2nd phosphorylation in the MAPK reaction, which can prevent plants from sensing extracellular stimuli in time and starting the intracellular immune defense mechanism. T3Es can inhibit the synthesis of lignin and diterpenoid phytochemicals to prevent plants from establishing their own physical barriers to resist the invasion of pathogenic bacteria and they can weaken plant hypersensitivity by inhibiting Ca^2+^-mediated ROS and NO synthesis. In order to cope with the invasion of pathogens, rice resistant near isogenic materials degraded hydrogen peroxide by double oxygenase at 24 h, thus reducing and eliminating the attack of reactive oxygen species on plant membrane lipids; At 48 h, it can resist the invasion of pathogenic bacteria by promoting lignin synthesis, disease-resistant protein production, monoterpene (chalcone) synthesis and indole alkaloid synthesis.

## 4. Materials and Methods

### 4.1. Experimental Materials

The near isogenic line NIL-*bls2* with BLS resistance in rice was used as the material. NIL-*bls2* was constructed in the BC_4_F_3_ generation derived from the crosses between susceptible indica rice variety ‘9311’ and Guangxi common wild rice material DY19, which contains BLS-resistant gene *bls2* [28]. The strains are Guangxi *Xoc* strain gx01 (abbreviated as WT) with high pathogenicity and its mutant strain (abbreviated as MT) with knockout of type III effector gene. The strains are provided by Professor Yongqiang He and Associate Professor Yanhua Yu, State Key Laboratory of Protection and Utilization of Subtropical Agricultural Biological Resources, respectively. The experiment was carried out at the scientific research base of the Agricultural College of Guangxi University, adopting the method of single-plant transplanting, planting 12 rows with 5 plants in each row, and row spacing at 15 cm × 25 cm. According to the field management method, we managed water, fertilizer, pests and diseases in the field (but did not manage BLS-related diseases).

### 4.2. Bacterial Inoculation and Identification of Diseased Spots

The strain was activated in NA medium at 28 °C for 48 h, and a single colony was inoculated in 250 mL of NB medium. After extended culture at 28 °C and 200 r/min for about 48 h to reach the logarithmic growth period, it was centrifuged at 5000 g/min for 5 min. The supernatant was discarded, and the cell precipitate was collected. The bacteria were resuspended and diluted to 3 × 10^8^ CFU/mL with sterile water for inoculation. Inoculation was carried out during continuous cloudy days at the tillering stage of rice, and acupuncture inoculation was carried out at a distance of about 5 cm from the leaf tip. Five leaves with the best growth from each plant were selected for inoculation. Among them, 3 leaves were inoculated 3 times per leaf, with the pinhole spacing of 1 cm for the transcription sequencing experiment; the remaining 2 leaves were inoculated once per leaf for measuring the length of diseased spots. Leaves used in the transcriptome experiment were collected at 12 h, 24 h and 48 h after inoculation. From each leaf of the three rice plants inoculated three times, the leaves of each inoculated site (about 1 cm) were cut and mixed as a biological replicate, and there were 3 biological replicates in total (Figure 7). The collected leaves were marked, quickly frozen with liquid nitrogen, and stored in a refrigerator at −80 °C. After 15 days of inoculation, the lengths of 4 diseased spots of each rice plant was measured. Compared with the four disease spots of the same rice plant, the spots with no signs of disease were removed, and the average value of the remaining disease spots was used for phenotypic identification. Differences among means were analyzed using a one-way ANOVA and Tukey’s HSD test. Statistical analyses were conducted in GraphPad Prism 7 software (GraphPad Software, La Jolla, CA, USA). Using the identification standard of resistance level of Nong Xiumei [35], the resistance level was identified according to the length of the disease spots. A lesion length ≤ 1.5 cm indicated disease resistance, and a lesion length > 1.5 cm indicated disease susceptibility.

### 4.3. RNA Sequencing

The collected leaves were taken out of the refrigerator at −80 °C, ground with liquid nitrogen, and the total RNA of leaves was extracted by TIANGEN RNA prep Pure Polysaccharide Polyphenol Plant Total RNA Extraction Kit (DP441, TIANGEN, Beijing, China). The concentration of RNA was detected using a Thermo Qubit 3.0 fluorescence quantitative analyzer (Thermo Fisher Scientific Inc., Waltham, MA, USA). The RIN value of RNA was determined using a Agilent 4200 Tape Station bioanalyzer (Agilent Technologies Inc., Palo Alto, CA, USA), and its integrity was evaluated. The purity of RNA was determined using a Thermo Scientific NanoDrop spectrophotometer ND8000 (Thermo Fisher Scientific Inc., Waltham, MA, USA). The qualified RNA was prepared with a VAHTS Universal V6 RNA-seq Library Prep Kit for Illumina (NR604, VAHTS, Nanjing, China) library building kit. After the library was constructed, an Agilent 4200 Tape Station bioanalyzer and an ABI Quant Studio 6 Flex Real-time PCR (Applied Biosystems Inc., Carlsbad, CA, USA) were used to control the fragment size and concentration. When the quality control of the library was qualified, it was sequenced by using the Illumina NovaSeq 6000 (Illumina, San Diego, CA, USA) sequencing platform PE150. The sequencing data yield of each sample was not less than 6G.

### 4.4. RNA Data Analysis

Experimental and control Raw Reads in sequencing data were filtered to remove Adapter sequences and low-quality sequences (the number of bases with a mass value ≤ 15 in reads is more than 40%) to obtain Clean Reads. The software hisat2 was used to compare the Clean Reads with the rice reference genome *Oryza sativa* Japonica Group, and the software was used to compare and analyze the Clean Reads of the reference genome. Transcripts of Mapped Reads are spliced and assembled by using software StringTie, and functional annotation is carried out through databases such as GO (Gene Ontology), KEGG (Kyoto encyclopedia of genes and genomes) and NR (NCBI non-redundant protein sequences). The R language package edgeR was used for gene differential expression analysis, and the DEGs screening threshold was FDR (false discovery rate) < 0.05, log_2_FC > 1 or log_2_FC < −1 (fold change, FC). The expression of the same gene between the experimental group and the control group at each time point of the different inoculation treatments was analyzed. After the genes with no expression difference between the experimental group and the control group were removed, the genes with different expressions between the two treatments and the control group (WT vs. H_2_O and MT vs. H_2_O) were merged at each time point. DEGs among different inoculation treatments in this gene set were analyzed, a Venn map and a volcano map were drawn. GO function annotation and KEGG enrichment analysis of DEGs were carried out by using the Shengxin analysis platform set up by Anhui Microanaly Genetech Co., Ltd. (Hefei, China). The software KOBAS was used to count the abundance of DEGs in the KEGG pathway and analyze the genes and fluxes related to plant disease resistance.

### 4.5. qRT-PCR Verification

The genes involved in key pathways in RNA-seq were screened and verified by qRT-PCR. Specific primers of candidate genes were designed by NCBI Primer BLAST (Table 1). Use VAHTS HiScript III RT Super Mix for qPCR (R323) kit, we added 1 μg RNA template, and performed a reverse transcription experiment according to the instructions. The rice actin gene was used as the internal reference gene, and three technical replications were set for each gene in each sample. We used a Roche lightcycler 480sybr green imaster kit to prepare 10 μL of a PCR reaction system according to the specification. The gene expression was detected by ABI Quant Studio 6 Flex Real-Time PCR. Reaction procedure: step 1: 95 °C/10 min; Step 2: 95 °C/15 s, 60 °C/1 min (collecting fluorescence), 40 cycles in total; Step 3: 72 °C/5 min; and Step 4 (dissolution curve): 95 °C/15 s, 60 °C/1 min, 0.05 °C/s, the temperature is raised to 95 °C (fluorescence collection), 95 °C/15 s. The relative expression of selected genes was calculated according to the 2^–ΔΔCT^ method.

## 5. Conclusions

In this study, through the transcriptome analysis of wild-type *Xoc* strain and its mutant strain lacking T3Es infecting rice resistance near isogenic lines, it was found that T3Es of pathogenic bacteria invaded the rice immune system mainly by inhibiting the signal transduction pathway mediated by ethylene, jasmonic acid, salicylic acid and the MAPK reaction, and inhibiting the synthesis of lignin and diterpenoid phytoalexins. The inhibitory effect was the strongest at 12 hpi, and gradually weakened at 24 hpi and 48 hpi. Rice NIL-*bls2* material can degrade hydrogen peroxide by dioxygenase, and promote the synthesis of lignin, disease-resistant protein, monoterpene antibacterial substances, indole alkaloids and other substances to resist the invasion of pathogenic bacteria. At the same time, 5 candidate genes *Os01g0719300*, *Os02g0513100*, *Os03g0122300*, *Os04g0301500* and *Os10g0575100* closely related to BLS were found. We provided new data resources and a theoretical basis for exploring the infection mechanism of *Xoc* strain gx01 and the resistance mechanism of resistance gene *bls2*.

## Figures and Tables

**Figure 1 ijms-24-00014-f001:**
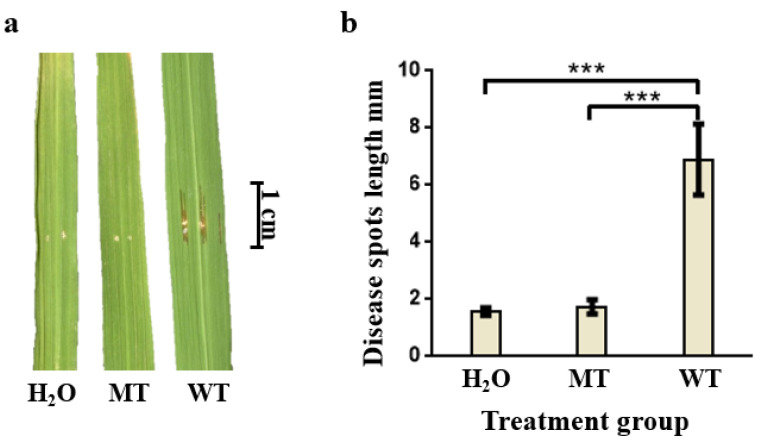
Investigation results of spots of WT, MT and H_2_O in three inoculation treatment groups. (**a**) Comparative phenotype of NIL-*bls2* leaves in three treatment groups. (**b**) Comparison of lesion lengths of WT, MT and H_2_O in inoculation groups. The correlation coefficients (Mean, SD and number) were H_2_O (1.54 mm, 0.14, 6), MT (1.72 mm, 0.26, 6), WT (6.87 mm, 1.24, 6). *** represents *p* < 0.001.

**Figure 2 ijms-24-00014-f002:**
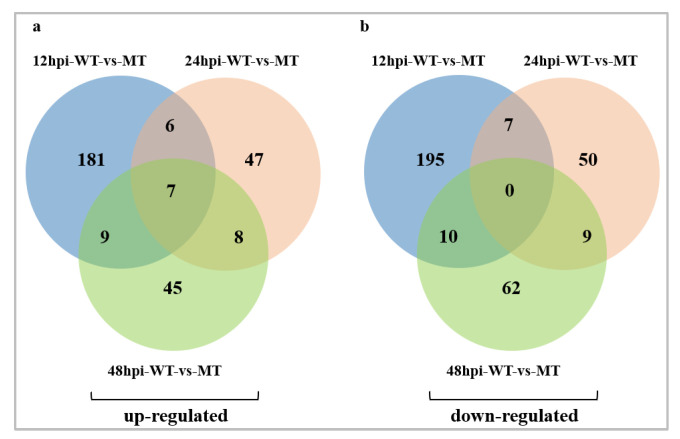
Venn diagram of DEGs distribution between WT-vs-MT at different time points. (**a**) The Venn diagram showing the overlapping up-regulated DEGs at different time points. (**b**) The Venn diagram showing the overlapping down-regulated DEGs at different time points.

**Figure 3 ijms-24-00014-f003:**
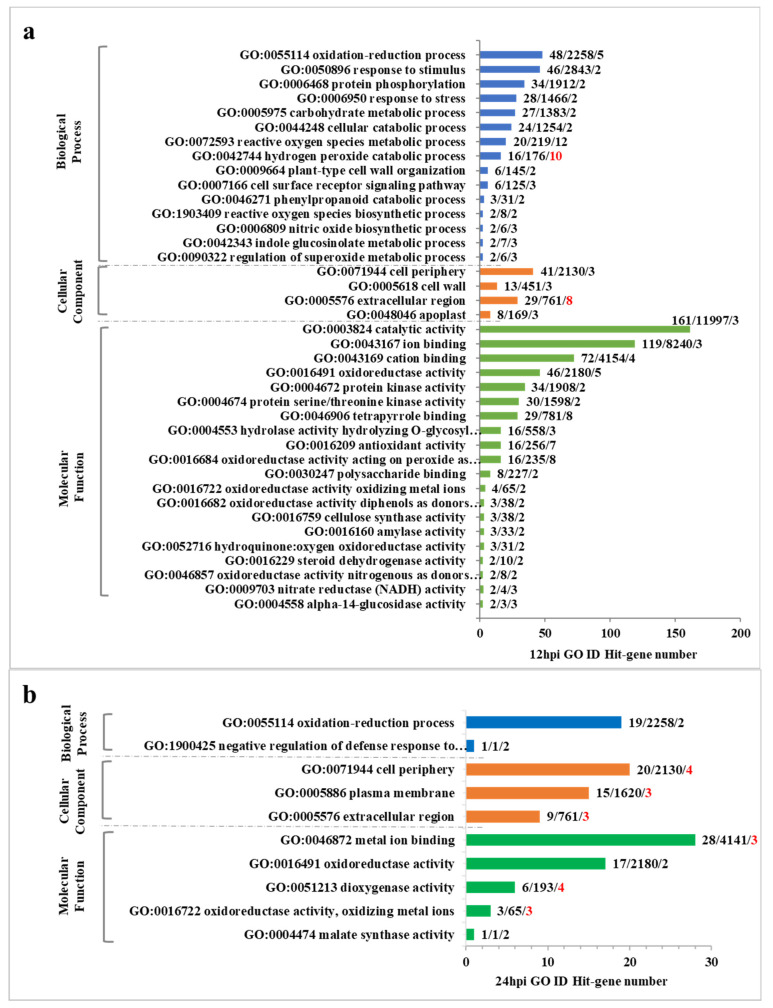
WT-vs-MT: GO enrichment significance analysis (*p* < 0.01). (**a**) 12 hpi GO enrichment significance analysis. (**b**) 24 hpi GO enrichment significance analysis. (**c**) 48 hpi GO enrichment significance analysis. The tag represents Hit_Gene number/background Gene number/−log10(*p*). The red font represents the GO term with prominent significance.

**Figure 4 ijms-24-00014-f004:**
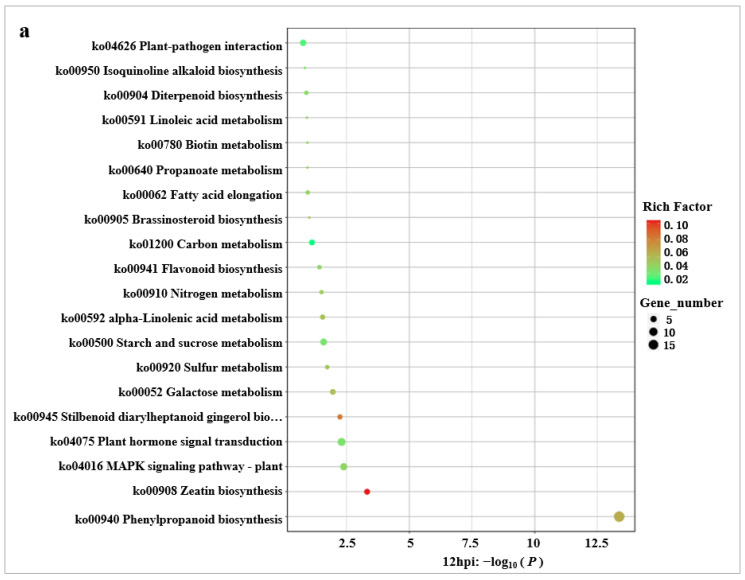
WT-vs-MT: KEGG enrichment significance analysis bubble chart (*p* < 0.01). (**a**) 12 hpi KEGG enrichment significance analysis bubble chart. (**b**) 24 hpi KEGG enrichment significance analysis bubble chart. (**c**) 48 hpi KEGG enrichment significance analysis bubble chart.

**Figure 5 ijms-24-00014-f005:**
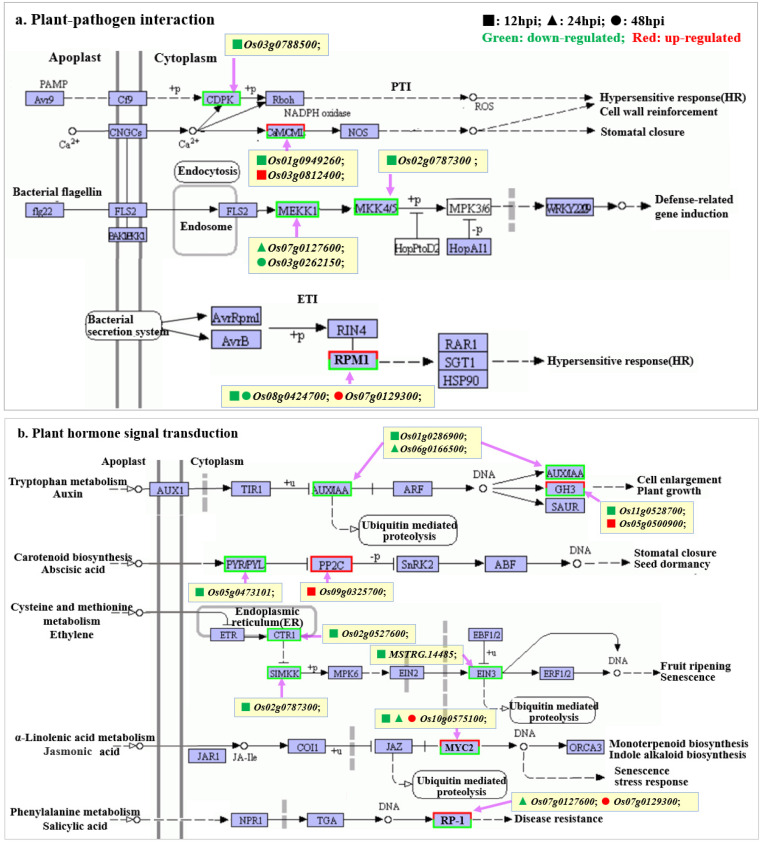
WT-vs-MT: Important KEGG pathways related to plant disease resistance mechanism. (**a**) Plant-pathogen interaction pathway; (**b**) Plant hormone signal transduction pathway; (**c**) MAPK signaling pathway-plant pathway; (**d**) α-Linolenic acid metabolism pathway; (**e**) Phenylpropanoid biosynthesis pathway; (**f**) Diterpenoid biosynthesis pathway; (**g**) Flavonoid biosynthesis pathway. ■: 12 hpi; ▲: 24 hpi; ●: 48 hpi; Green: down-regulated; Red: up-regulated.

**Figure 6 ijms-24-00014-f006:**
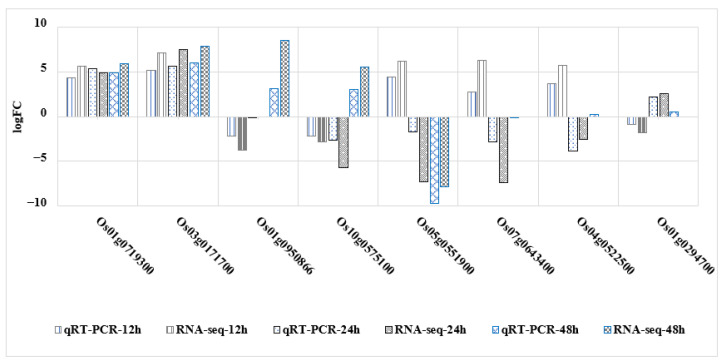
WT-vs-MT qRT-PCR validation results and RNA-seq sequencing results.

**Figure 7 ijms-24-00014-f007:**
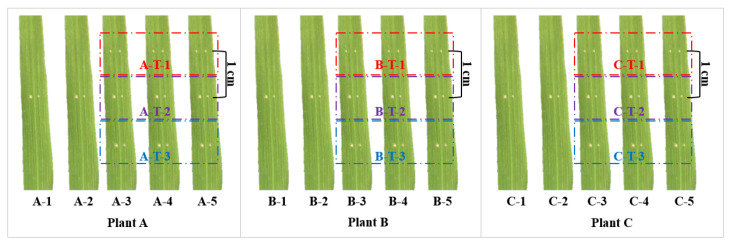
Schematic diagram of rice inoculation and sampling. Three rice plants A, B and C were inoculated with 5 leaves each, leaves 1, 2 were inoculated once for the measurement of diseased spots; leaves 3–5 were used for transcriptome sequencing, and each leaf was inoculated 3 times, and the inoculation interval was 1 cm. The T-1 region of the leaves of three rice plants was collected as biological repeat 1 and the T-2 region and T-3 region as biological repeat 2 and 3, respectively.

**Table 1 ijms-24-00014-t001:** qRT-PCR primer sequence.

Gene Name	Forward Primer Sequence	Reverse Primer Sequence
*Os01g0719300*	CTAGCACTGGCAGAAGGAATAG	CATGATACCAAACGCGATCATC
*Os01g0294700*	ACAACACCTACTACCACAACAA	TACAAGCATAGTCGTCGTTGAT
*Os01g0950866*	GTATTGATCCCTCCAGATGCGT	GCTTGCTCAAAGTCACCAGACA
*Os03g0171700*	TCTCCTGAAAAGGGAAAAGTGT	TCGATCTCTATTAGCCCTGGTA
*Os04g0522500*	CCTCCCAAACTTTTTCAAGCAT	TCGAGTTCACATCGACCATAAA
*Os05g0551900*	CATCAAGCCCCACAGAGCA	CCCACCAATCCACAAAACG
*Os07g0643400*	GATCCACTCCTTCTACGTCTTC	AATTAAGCATTCGATTTGGGCG
*Os10g0575100*	CCATCCTAACACAGATCTTT	ATACTCTCTACTCTCCCCAA

## Data Availability

All data generated or analysed during this study are included in this published article and its supplemental tables.

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
