# Peer review of "Transcriptome Analysis in Response to Infection of Xanthomonas oryzae pv. oryzicola Strains with Different Pathogenicity"

_ijms, 2022, doi:10.3390/ijms24010014_

Round 1

Reviewer 1 Report

In the future, it is necessary to  include more genotypes of the rice and of the  pathogen in order to develop possible resistance and/or tolerance strategies. It is also necessary to take into account all forms of interaction in future analysis.

Author Response

We totally agree with the reviewer:“In the future, it is necessary to  include more genotypes of the rice and of the  pathogen in order to develop possible resistance and/or tolerance strategies. It is also necessary to take into account all forms of interaction in future analysis.” Related research work is also being gradually advanced.

Reviewer 2 Report

The manuscript inoculated NIL-bls2 plant with mock, WT Xoc and T3Es deleted MT mutant. Then transcriptome RNA sequences were analyzed through a series of bioinformatical technology including DEGs, GO, KEGG etc. In fact, the authors overemphasized the the bioinformatical analysis of RNA sequences, especially among them at different HPIs. Although Water was taken as mock treatment, the water treatment has never been employed in the subsequent analysis. In addition, the confirmation of RNA seq using qRT-PCR is too thin to provide solid support owing of too little DEGs number. 

Others detail comments: 

1.     About the 7 overlapping DEGs among different time points, no further introduction or study in following works. Then, what is the main objects to display the overlapping DEGs?

2.     Fig 3b, MT-48hpi-3

3.     What are the relationship between seven overlapping DEGs in Fig 2 and the four closely related genes to BLS in Table 1? Is there any duplicate? Considering to the four closely related genes, did any documents report their biological function on disease resistance in rice?

4.     It is clearer and more readable to use simple annotation instead of the GO term directly in Fig 4. To modify “namuber”.

5.     Fig 7, it is better to use the original expression patterns of each gene of each treatment in qRT-PCR and RNA sequence although it will greatly raise the number of sample data.

6.     In Abstract, the 1st and 2nd phosphorylation in the MAPK cascade. To use the phosphorylation of MAPKKK and MAPKK.

7.     In abstract, “rice NIL-bls2 material can promote wound healing by promoting the synthesis 30 of traumatic acid at 12h”, is there any experimental proof to support it?

8.     Rewrite “A recessive bacterial streak resistance gene bls1 has been cloned, which 83 encodes a mitogen-activated protein kinase (OsMAPK6). Over-expression of BLS1 or low-84 expression of bls1 can weaken or enhance the resistance of rice to Xoc strain JZ-8, respec-85 tively.” In line 83-85.

9.     Oryza sativa (Italic) in Line 494; KEGG in Line 497

10.   In abbreviations: pv. oryzicola (Italic)

Author Response

Response to Reviewer 2 Comments

Point:The manuscript inoculated NIL-bls2 plant with mock, WT Xoc and T3Es deleted MT mutant. Then transcriptome RNA sequences were analyzed through a series of bioinformatical technology including DEGs, GO, KEGG etc. In fact, the authors overemphasized the the bioinformatical analysis of RNA sequences, especially among them at different HPIs. Although Water was taken as mock treatment, the water treatment has never been employed in the subsequent analysis. In addition, the confirmation of RNA seq using qRT-PCR is too thin to provide solid support owing of too little DEGs number. 

Response:

In this study, H2O was the control group (lines 102-103 in this paper), and WT and MT were the experimental groups.“After removing the genes with no expression difference between the experimental group and the control group at each time point of each material, thus excluding the genes unrelated to BLS resistance response, the DEGs of rice materials in WT group and MT group at different time points after inoculation were statistically analyzed to reveal the expression patterns of resistance-related genes”(lines 122-126 in this paper). In short, this study firstly analyzed the gene expression difference of WT vs. H2Oand MT vs.H2O, and removed the genes with no expression difference in MT vs. H2O and WT vs. H2O. Secondly, the DEGs between WT and MT was selected from the remaining genes (lines 502-506 in this paper). So all the DEGs screened were based on comparison between the experimental group and the control group (H2O), the water treatment has been employed in the subsequent analysis.

As there are too many DEGs in this study, we randomly selected six genes. The qRT-PCR results are basically consistent with the expression trend of RNA sequencing results, which proves that the analysis based on RNA sequencing data is true and reliable. Although the number of genes verified is small, the results are convincing.

Others detail comments: 

Point 1: About the 7 overlapping DEGs among different time points, no further introduction or study in following works. Then, what is the main objects to display the overlapping DEGs?

Response 1: The overlapping areas in Fig. 2 only show the distribution of DEGs related to resistance between WT and MT identified. The 7 overlapping DEGs are Os01g0719300, Os03g0171700, Os03g0259100, Os04g0286333, Os04g0581000, Os06g0678800, MSTRG.3500 (red font content in Supplemental Table 2). The follow-up research related to resistance is mainly based on the gene annotation, GO and KEGG analysis of identified DEGs. These analysis results are related to the 7 overlapping DEGs. For example, gene Os01g0719300 is closely related to BLS (Table 1), and Os03g0171700 is annotated as bHLH transcription factor (Supplemental Table 3).

Point 2: Fig 3b, MT-48hpi-3

Response 2: The related content has been modified.

Point 3: What are the relationship between seven overlapping DEGs in Fig 2 and the four closely related genes to BLS in Table 1? Is there any duplicate? Considering to the four closely related genes, did any documents report their biological function on disease resistance in rice?

Response 3: seven overlapping DEGs in Fig 2 had significant expression differences(log2FC>1 or log2FC<-1)at all three time points. While four DEGs in table 1 had highly significant expression differences(log2FC>2 or log2FC<-2)at least one time points. Among them, and Os01g0719300 is DEGs located in the overlapping areas of Fig. 2. Moreover,It have been reported that this four genes were closely related to disease resistance in rice: Os01g0719300(OsSULTR3:6) has been reported as a BLS-sensitive gene[28];Os02g0513100(OsSWEET15) has been proved to be the TALEs target secreted by Xan-thomonas oryzae pv. oryzae (Xoo) [29,30];in resting cells, JAZ protein acts as a JA transcription inhibitor by binding with MYC2 positive transcription regulator [31],Os10g0575100 as a MYC2-related gene;In the nucleus, OsbHLH6(Os04g0301500)interacts with OsMYC2, which is a key regulatory protein in JA signaling pathway;OsbHLH6 aslo can prevent OsJAZ from inhibiting the transcription of OsMYC2, which is a negative regulatory protein of JA signal, thus promoting the activation of JA signal [32]. (lines 390-417 in this paper).

Point 4: It is clearer and more readable to use simple annotation instead of the GO term directly in Fig 4. To modify “namuber”.

Response 4: Because some GO IDs have similar comments, if simple comments are used, GO IDs need to be displayed at the same time to express clearly. However, there are some problems, such as too long annotations on the ordinate of the figure, incomplete presentation of some annotations, and too much space occupied by the picture in the paper. Therefore, we choose the ordinate to show only GO ID, and explain the ID in the explanatory text.

To modify “namuber”: The related content has been modified.

Point 5: Fig 7, it is better to use the original expression patterns of each gene of each treatment in qRT-PCR and RNA sequence although it will greatly raise the number of sample data.

Response 5: Using the original expression pattern of each gene in each treatment can show the expression trend of qRT-PCR and RNA sequence more truly and fully, but the picture display will become complicated and unclear. The display mode in Figure 7 will be more intuitive and clear in comparing the expression trends of qRT-PCR and RNA sequences.

Point 6: In Abstract, the 1st and 2nd phosphorylation in the MAPK cascade. To use the phosphorylation of MAPKKK and MAPKK.

Response 6: The contents in the abstract and abbreviations have been modified.

The modified content in the abstract:the MAPKK(MAPK kinase) and MAPKKK(MAPK kinase kinase) in the MAPK(mitogen-activated protein kinase) cascade reaction

Add content  In abbreviations:MAPKK: MAPK kinase; MAPKKK: MAPK kinase kinase;

Point 7: In abstract, “rice NIL-bls2 material can promote wound healing by promoting the synthesis 30 of traumatic acid at 12h”, is there any experimental proof to support it?

Response 7: In line 354-357. “Among the metabolic pathways of α -linolenic acid (Fig.6 d), the genes related to traumatic acid synthesis pathway in WT group samples at 12hpi were up-regulated, indicating that plants were repairing the inoculation wounds at this time.”

Point 8: Rewrite “A recessive bacterial streak resistance gene bls1 has been cloned, which encodes a mitogen-activated protein kinase (OsMAPK6). Over-expression of BLS1 or low-84 expression of bls1 can weaken or enhance the resistance of rice to Xoc strain JZ-8, respectively.However, over-expression of BLS1 or low-expression of bls1 can improve the broad-spectrum resistance of rice[21].” In line 83-87.

Response 8: Rewrite as follows:

Recently, a recessive resistance gene bls1 has been cloned, which encodes a mitogen-activated protein kinase (OsMAPK6). Both overexpression of BLS1 and low expression of bls1 could increase of salicylic acid and induce expression of defense-related genes, simultaneously increasing the non-race specific broad-spectrum resistance. Moreover, low expression of bls1 showed increase of JA and ABA (abscisic acid), in company with an increase the race specific resistance to Xoc strain JZ-8[21].

Point 9: Oryza sativa (Italic) in Line 494; KEGG in Line 497

Response 9: The related content has been modified.

Point 10:  In abbreviations: pv. oryzicola (Italic)

Response 10: The related content has been modified.

Reviewer 3 Report

The transcriptome-based article reveals nice information that seems to be beneficial for host-pathogen interaction studies. Xoc strains gx01 and its respective mutant strains were employed to infect the rice NILs to reveal the differential expression genes (DEGs) at different time points. The authors have elaborated the article in a great manner; on the contrary, there are some minor suggestions that needs to be improved:

-This article needs minor English editing improvements.

-All the Figures should have good quality owing to the fact that many figures are blurred and do not have clear visualization. Please improve them.

-Line 123: Please rewrite the statement.

-Line 130: Please rewrite the statement. Too long sentence.

-Please rewrite the GO terms analysis results in terms of the proper scientific way in order to make a sense of your results.

-Please retrieve the transcription factors, involved in the resistance and susceptibility conditions.

-Line 370: how authors can say that given upregulated genes are related to resistance mechanism, why not down-regulated ones? I hereby inform you that genes are linked to each other and their coexpression regulation pattern determines the resistance and susceptibility mechanism. Hope you will change your statement accordingly with proper reference.

Author Response

Point 1: -This article needs minor English editing improvements.

Response 1: The related content has been revised in the paper.

Point 2: -All the Figures should have good quality owing to the fact that many figures are blurred and do not have clear visualization. Please improve them.

Response 2: The figures have been improved in the paper and clear pictures are also stored in attachment Fig1-7(PPT).

Point 3: -Line 123: Please rewrite the statement.

Response 3: The related content has been revised in the paper.

Point 4: -Line 130: Please rewrite the statement. Too long sentence.

Response 4: The related content has been revised in the paper.

Point 5: -Please rewrite the GO terms analysis results in terms of the proper scientific way in order to make a sense of your results.

Response 5: The related content has been revised in the paper.

Point 6: -Please retrieve the transcription factors, involved in the resistance and susceptibility conditions.

Response 6: Line 141: In DEGs, we retrieved 10 transcription factor-related genes. The specific information is in Supplementary Table 3.

Point 7: -Line 370: how authors can say that given upregulated genes are related to resistance mechanism, why not down-regulated ones? I hereby inform you that genes are linked to each other and their coexpression regulation pattern determines the resistance and susceptibility mechanism. Hope you will change your statement accordingly with proper reference.

Response 7: We quite agree with the expert's point of view:genes are linked to each other and their coexpression regulation pattern determines the resistance and susceptibility mechanism. There is a problem of inappropriate language expression in Line 369, which has been modified.
